# Adding a Leafy Vegetable Fraction to Diets Decreases the Risk of Red Meat Mortality in MASLD Subjects: Results from the MICOL Cohort

**DOI:** 10.3390/nu16081207

**Published:** 2024-04-18

**Authors:** Rossella Donghia, Rossella Tatoli, Angelo Campanella, Francesco Cuccaro, Caterina Bonfiglio, Gianluigi Giannelli

**Affiliations:** 1National Institute of Gastroenterology—IRCCS “Saverio de Bellis”, 70013 Castellana Grotte, Italy; rossella.tatoli@irccsdebellis.it (R.T.); angelo.campanella@irccsdebellis.it (A.C.); catia.bonfiglio@irccsdebellis.it (C.B.); gianluigi.giannelli@irccsdebellis.it (G.G.); 2Local Health Unit—Barletta-Andria-Trani, 76121 Barletta, Italy; francescocuccaroepi@gmail.com

**Keywords:** mortality, red meat, vegetables

## Abstract

Background: Dietary guidelines recommend limiting red meat intake because it has been amply associated with increased cancer mortality, particularly in patients with liver conditions, such as metabolic dysfunction-associated fatty liver disease (MASLD). MASLD is the leading cause of liver dysfunction in the world today, and no specific treatment other than lifestyle correction has yet been established. The aim of this study was to explore the protective role of leafy vegetables when associated with high red meat consumption. Methods: The study cohort included 1646 participants assessed during the fourth recall of the MICOL study, subdivided into two groups based on red meat intake (≤50 g/die vs. >50 g/die), in order to conduct a cancer mortality analysis. The prevalence of subjects that consumed >50 g/die was only 15.73%. Leafy vegetable intake was categorized based on median g/die consumption, and it was combined with red meat intake. Conclusions: This is the first study to demonstrate that the consumption of about 30 g/die of leafy vegetables reduces the risk of mortality. A strong association with mortality was observed in subjects with MASLD, and the protective role of vegetables was demonstrated.

## 1. Introduction

The American Cancer Society (ACS) and the International Agency for Research on Cancer (IARC) have reported that worldwide, one in five people develop cancer in their lifetime. According to 2023 estimates, the most frequent forms are those of breast, colorectum, bladder, lung, and prostate [www.airc.it, accessed on 10 January 2024]. Lifestyle factors such as diet, alcohol intake, smoking, physical activity, and weight are modifiable risk factors for the development of cancer [1].

Cancer therapy is increasingly complex, with the role of diet, nutrition, and physical activity as adjuvants to therapy, with the goal of improving treatment efficacy and reducing the acute and long-term toxicity of surgery, chemotherapy, and radiation. Diet and lifestyle interventions should be integrated into health care and cancer survivorship programs to provide survivors with effective guidelines and monitoring strategies [2].

The International Agency for Research on Cancer (WHO-IARC) recommends a diet with a rich content of fruits, vegetables, legumes, and whole grains, in addition to limited consumption of red meat, processed foods, and sugary drinks [www.wcrf.org, accessed on 10 January 2024].

The WHO-IARC classified red meat as probably carcinogenic in humans (Group 2A) [3]. The World Cancer Research Fund and the American Institute for Cancer Research (WCRF/AICR) recommend a consumption of less than 500 g/week of red meat [1], because it is positively associated with prostate, pancreatic, and colorectal cancer [4]. The IARC working group estimated that for every additional 100 g of red meat consumed, the relative risk of colorectal cancer increases by about 18%, as compared to that in individuals who eat the least meat [3].

Red meat is defined as unprocessed muscle meats, including beef, veal, pork, lamb, mutton, horse, or goat meat [3]. Experimental studies have identified several components of red meat that promote carcinogenesis, many of which are formed during the cooking or the digestion of meat [5,6,7,8]. N-nitrose compounds (NOCs) are formed in the digestive tract [9], while heterocyclic aromatic as amines (HAAs) are formed in well-done cooked meats and poultry [10,11]. Polycyclic aromatic hydrocarbons (PAHs) are formed in smoked meats and meats cooked under the flame [12]. All these chemical components may exert a pro-carcinogenic effect because they cause direct DNA damage to the double helix [13]; in fact, during cell division, mutations may occur leading to tumor development [14]. Furthermore, ingested heme can catalyze the nitrosation of endogenous secondary amines [15] and exert pro-oxidative effects through catalysis of lipid peroxidation in the gastrointestinal tract [16], contributing to inflammation and tumor promotion.

Despite the evidence, moderate consumption of red meat, as a part of a healthy and balanced diet, does not lead to health problems [17]. In fact, red meat is an important source of protein, because it contains all essential amino acids, and also highly bioavailable iron, zinc, selenium, and B vitamins, especially vitamin B12 [18]. Changing diet, eating habits, and cooking methods can mitigate some of the chemical exposures that contribute to cancer risk [19].

For a balanced diet, the daily consumption of vegetables is also among the 10 recommendations for cancer prevention compiled by the AIRC-WCRF [16]. A diet rich in plant-based foods is associated with a reduction in the risk of developing chronic degenerative diseases including cancers, due to the presence of several nutrients and phytochemicals with protective, anti-inflammatory, and anti-oxidant effects [20,21,22]. A recent meta-analysis demonstrated an inverse association between fruit and vegetable intake and the risk of cancer and all-cause mortality [23]. In particular, for total cancer risk, an inverse association was observed with the consumption of green–yellow vegetables and cruciferous vegetables [23]. However, further studies are needed to evaluate the effects of different types of vegetables. 

Adherence to a healthy and balanced diet is also one of the main tools for the prevention and management of metabolic dysfunction-associated fatty liver disease (MASLD) [24]. MASLD, formerly known as non-alcoholic fatty liver disease (NAFLD), is the leading cause of liver dysfunction in the world today. There is currently no specific treatment for this condition other than lifestyle correction [25]. A recent study showed that a healthy diet, based on the consumption of fruits, vegetables, and nuts, as well as important sources of vitamin A, vitamin C, pyridoxine, and potassium, is inversely associated with the risk of MASLD [26]. 

The present cohort study aims to evaluate the protective role of vegetables and their interaction with red meat in preventing cancer mortality in a cohort resident in Southern Italy.

## 2. Materials and Methods

### 2.1. Study Design and Population

The study involved 1646 subjects derived from a cohort of the Multicenter Italian study on Cholelithiasis (MICOL), approved by the Minister of Health. The MICOL study is a systematic study of a random sample (draw step of 5) from the electoral rolls of Castellana Grotte. The study began in 1985 and continued in 1992, 2005–2006, and 2013–2016. In 2005–2006, using the same sampling scheme, this cohort was supplemented with a random sample of subjects (called “PANEL Study”) aged 30–50 years to compensate for the aging of the cohort. MICOL recruitment for this study started in 2017 (2nd recall, called MICOL 3), (Figure 1) [27].

Methodological details of this population-based study have been described in previously published papers [28,29]. The study was conducted in accordance with the Helsinki Declaration of 1975 and adheres to the “Standards for Reporting Diagnostic Accuracy Studies” (STARD) guidelines. (http://www.stard-statement.org/, accessed on 5 October 2019). The manuscript was organized according to the “Strengthening the Reporting of Observational Studies in Epidemiology-Nutritional Epidemiology” (STROBE-nut) guidelines (https://www.strobe-nut.org/, accessed on 5 October 2019), and participants signed informed consent forms before undergoing examination.

The study was approved by the Ethics Committee of the National Institute of Gastroenterology and Research Hospital (DDG-CE-589/2004, 18 November 2004).

Red meat (Appendix A) consumption (g/die) was categorized based on data from the World Cancer Research Fund International at 50 g/die, and leafy vegetable intake was based on the median values of our distribution (>28.49 g/die vs. ≤28.49 g/die). These food groups were created based on previous papers published in the literature [30].

Mortality rates for malignant cancers were updated on 31 December 2023 and diseases were classified based on ICD-10, considering cancers of the lip, oral cavity, pharynx, skin and soft tissue, digestive system, respiratory system and intrathoracic organs, breast, female and male genital organs, urinary system, lymphatic tissue, and hematopoietic tissue and related tissues, as well as some other tumors (Appendix A).

### 2.2. Lifestyle, Clinical, and Dietary Assessment

Lifestyle and anthropometric assessments were performed by a physician during examination at the study center. Smoking status was based on the single question, “Do you smoke?” The level of education was expressed as years of schooling. 

Body mass index (BMI) was calculated as kg/m^2^. Height and weight measurements were performed using a Seca (Model: 220) stadiometer and a Seca (Model: 711) scale.

Blood was collected from all subjects in the morning after an overnight fast. After overnight fasting, blood samples were collected between the hours of 8:00 and 9:00 a.m. Serum was examined for glucose, as well as lipid profile (total cholesterol, HDL, LDL, triglycerides, and fatty acids). Glucose concentrations were determined using the glucose oxidase method, and lipid concentrations (total cholesterol, HDL, LDL, triglycerides, and fatty acids) were measured using an automated colorimetric method (Sclavus, Siena, Italy) (Hitachi; Boehringer Mannheim, Mannheim, Germany). An automated system measured total bilirubin, GOT, GGT, and SGPT, in accordance with standard laboratory procedures (UniCel Integrated Workstations DxC 660i, Beckman Coulter, Fullerton, CA, USA). 

The serum was separated into aliquots. One aliquot was immediately stored at −80 °C. The second aliquot was used to test biomarkers through standard laboratory techniques in our laboratory. Other aliquots were stored in the biobank according to validated protocols, and processed by expert personnel.

Diet and eating habits were investigated by administering a previously validated food frequency questionnaire, administered during the visit. The questionnaire is organized into 11 sections representing food macro areas: grains, meat, fish, milk and dairy products, vegetables, legumes, fruits, various foods, water and alcoholic beverages, olive oil and other edible fats, coffee/sugar, and salt. Subsequently, each food (86 foods in total) was converted to mean daily intake in grams, as previously carried out for other studies [29].

### 2.3. Statistical Analysis

Patients’ characteristics are reported as mean and standard deviation (M ± SD) for continuous variables, and as frequency and percentage (%) for categorical variables. To test the association between the independent groups (Red Meat > 50 g/die vs. Red Meat ≤ 50 g/die), a chi-squared test was used for categorical variables, while the Wilcoxon rank-sum test (Mann–Whitney) was used to compare parameters between two independent groups. 

The associations between red meat consumption, interactions between red meat and leafy vegetable intake, and the risk of mortality were assessed using Cox’s proportional hazards regression model, and the results were presented as Hazard Ratio (HR) with 95% confidence intervals (95% C.I.). Models were adjusted for covariates (age, gender, BMI, cholesterol, triglycerides, and Kcal). Survival probability was explored using the non-parametric Kaplan–Meier method, and the equality of survival curves was analyzed with the log-rank test.

To test the null hypothesis of non-association, the two-tailed probability level was set at *p*-value = 0.05 (probability of the null hypothesis, i.e., the idea that a theory being tested is false). The analyses were conducted with Stata Statistical Software: Release 18, StataCorp, 2023, StataCorp LLC.: College Station, TX, USA.

## 3. Results

Cohort characteristics stratified by red meat intake are shown in Table 1.

Males were more prevalent among subjects who ate >50 g/die red meat (76.83% vs. 51.41%, *p* < 0.001) and those in this category were younger (61.84 ± 8.88 vs. 66.06 ± 9.43 years, *p* < 0.0001), showing statistically significant differences. Smokers showed the same trend as major red meat consumers (22.39% vs. 13.02%, *p* < 0.001), again showing statistically significant differences. Systolic pressure was lower in the second group (136.30 ± 95.37 vs. 139.12 ± 82.62, *p* < 0.0001) and hypertension was lower in subjects who ate less red meat (39.15% vs. 51.16%, *p* < 0.001). Lower levels of blood parameters were observed in the first group, for HDL (87.71 ± 618.38 vs. 58.60 ± 267.44, *p* = 0.01), triglycerides (200.22 ± 627.46 vs. 142.39 ± 279.00, *p* = 0.002), and GGT (174.75 ± 1233.01 vs. 82.94 ± 801.90, *p* = 0.0002), all of which were significantly different. Vice versa, lower levels of SGPT were observed in the second group (18.83 ± 14.17 vs. 25.12 ± 268.64, *p* = 0.0002). Leafy vegetable intake was associated with red meat intake, with a higher intake in subjects consuming > 50 g/die (56.76% vs. 48.09%, *p* = 0.01). 

The association of red meat consumption and its relative interaction with leafy vegetables is shown in Table 2.

Higher grams of consumption of red meat show a positive association with the highest risk of mortality (HR = 2.02, *p* = 0.001, 1.36 to 3.01 95% C.I.), but a statistically significant trend is observed for the interaction between red meat and leafy vegetables. In the category with a higher red meat consumption combined with a lower or high leafy vegetable intake, a reduction of risk was observed, which was statistically significant (HR = 2.36, *p* = 0.002, 1.38 to 4.01 95% C.I.; HR = 1.85, *p* = 0.03, 1.06 to 3.22 95% C.I). Kaplan–Meir survival probability analyses revealed that a high consumption of red meat is associated with a decreased survival (or risk increase) compared to lower levels of this food (Appendix A and Figure 2, respectively), and the difference between the curves is statistically significant (*p* = 0.05).

The survival probability analysis of the combination of red meat and leafy vegetable intake is shown in Figure 3, with a higher probability of survival at the last level (yellow line) compared to less leafy vegetable consumption (green line).

## 4. Discussion

Epidemiological evidence shows that high consumption of red meat increases the risk of developing cancer, while the consumption of fruit and vegetables has been shown to have an opposite, as well as protective, role. 

In this study, the consumption of approximately 30 g/die of leafy vegetables reduced the risk of mortality by about 22%, and a strong association with mortality was observed in subjects with MASLD, although it is a cohort that adheres very closely to the Mediterranean diet, because it consumes little meat. 

Many papers in the literature have been based on the replacement of red meat with leafy vegetables [31], but no studies based on the combination of vegetables with red meat have previously been conducted in a specific cohort from Southern Italy. The benefits were probably due to changes in the intake of nutrients, such as fatty acids, proteins, heme iron, sodium, dietary fiber, minerals, phytochemicals, and other bioactive components [31] typical of plants.

Several mechanisms may explain the association between MASLD and meat consumption, including hepatic lipid accumulation, hepatic insulin resistance, and oxidative stress [32,33]. A cross-sectional study, conducted using 2198 subjects, identified a positive relationship between red meat consumption and GGT [33]. GGT, in addition to being a biomarker of fat hepatic accumulation and insulin resistance [34], may be considered a nonspecific indicator of oxidative stress, leading to the hypothesis of a role of oxidative stress in the development of chronic diseases related to meat consumption [33]. The results reported by Fan N et al. also support the positive associations between GGT, ALT, and TG and red meat consumption [35]. 

Further studies support the association between meat consumption and risk of MASLD [36,37]. In contrast to previous studies [38], Wen G et al. found no correlation between higher vegetable consumption and risk of MASLD [39]. This result is probably explained by the different fiber and phytochemical contents of vegetables [40]. In fact, it is vegetables with higher fiber content that are associated with reduced risk of MASLD [38].

Red meat (such as beef, pork, or lamb) and processed meat intakes have been demonstrated to increase the risk of cancer, but also of atherosclerosis, type 2 diabetes, and all-cause mortality [41,42,43,44], as well as the risk of endothelial dysfunctions [45]. Furthermore, they have been shown to be associated with increases in fecal water genotoxicity due to higher levels of DNA damage in the human colonic epithelium, inducing the carcinogenic process. The heme protein is probably the main cause of this increase, because it would induce oxidative damage [46].

From a physiological perspective, a meat-rich diet has several potential nutritional benefits, but also some potential adverse properties. Meat is rich in protein, iron, zinc, and B vitamins, as well as vitamin A, all of which are very bioavailable in meat compared to plant products such as grains and leafy greens.

A recent meta-analysis indicated an approximately 20% higher risk of colorectal cancer per 100 g/day increased consumption of red meat and 50 g/day increased intake of processed meat. The risk increases linearly with the increasing intake of red and processed meats up to approximately 140 g/day; beyond this level, the risk increase is less pronounced [47]. Several mechanisms could underlie the relationship between red meat consumption and cancer. Meat is an abundant source of sulfur-containing amino acids and saturated fats, and in processed meat, inorganic sulfur is used as a preservative. Heme iron in red meat could induce oxidative stress, colonocyte proliferation, and the endogenous formation of N-nitrose compounds (NOCs), which are potent carcinogens in the gastrointestinal tract. Meat cooked at a high temperature or flame is also a source of other mutagens, including heterocyclic amines (HCAs) and polycyclic aromatic hydrocarbons (PAHs). High consumption of heme iron (but not other forms of iron), NOCs, HCAs, and PAHs have all been associated with an increased risk of colorectal tumors. Processed meat and heme iron have been more strongly associated with the risk of colorectal neoplasia where mutations in KRAS and APC were identified [48]. 

Meats also contain protecting factors such as carotenoids, flavonoids, and phytochemicals—although these are also present in fruit and vegetables—and derived factors such as folic acid, selenium, zinc, and other components [49].

The protective role of fruit and vegetables against the risk of developing tumors could derive from their high contents of fiber and polyphenols [47]. The health-promoting effects could be linked to the gastrointestinal ability to increase fecal volume and reduce transit, allowing a decreased fermentation process in the colonic lumen with evident anti-tumor effects [50,51]. The protective effects of polyphenols have been linked to their anti-inflammatory, antioxidant, and pro-apoptotic characteristics [52].

Fruits and vegetables could protect against cancer because they have high levels of several potential anticarcinogenic compounds, such as folates, fibers, B vitamins, antioxidants, and minerals. Dietary fiber has been shown to decrease levels of deoxycholic acid (DCA), a bile acid produced via the dehydroxylation of primary bile acids that is known to be a direct tumor promoter in the development of cancer [53,54,55].

However, epidemiologic studies have yielded inconsistent findings, possibly because of the large variations across studies in terms of consumption amounts, production methods, storage conditions, nutrient contents, and cooking and preparation, as well as variations accounting for confounding factors. In 11 of 21 cohort studies, a weak inverse association was reported between fruit or vegetable intake and colon cancer, while in other studies no association was detected [56]. 

It is clear that decreasing the amount of red meat in the diet will limit the level of oxidative catalysts in the tissues. On the contrary, it is recommended to increase the consumption of fruit and vegetables because this increases the levels of antioxidant components, such as selenium, vitamin E, vitamin C, lycopene, cysteine-glutathione, and various phytochemicals. 

Decreasing the amount of red meat in the diet and increasing antioxidants in the diet are the main protections against oxidation catalyzed by heme. The detrimental processes involving the catalysis of heme oxidation are not well recognized. Raw red meat contains high levels of oxyhemoglobin on the surface of the muscle, and deoxymyoglobin and deoxyhemoglobin inside. In cooked red meat, myoglobins, hemoglobins, and cytochromes are converted into denatured heme proteins, hemichromes, and hemochromes [57].

## 5. Conclusions

In conclusion, in the present study we observed a significant association between the consumption of red meat, together with leafy vegetables, and cancer mortality outcome. These findings may indicate that limiting the intake of red meat should be encouraged, but that eating it together with vegetables can mitigate the worst health outcomes, including in MASLD conditions.

In summary, the interaction between vegetable fibers and red meat involves various physiological processes, including digestion, satiety, blood sugar regulation, and gut microbiota modulation. Incorporating fiber-rich vegetables into meals containing red meat can have several potential health benefits, including improved digestive health, enhanced satiety, better blood sugar control, and support for healthy gut microbiota. However, individual responses may vary, and dietary recommendations associated with physical activity should be tailored to specific health goals and considerations.

The balance of risk-promoting and protecting factors within the diet is important in public health, to protect against diseases, and to reestablish the balance between healthy and disease conditions.

Patients should be encouraged to eat a balanced and varied diet, paying close attention to food quality and quantity. In fact, they should be taught to reduce the intake of saturated fat and cholesterol associated with red meat consumption, and include a variety of vegetables vitamins, minerals, antioxidants, and phytochemicals in their meals.

### Strengths and Limitations

The strengths of the present study include the large cohort and the generalizability of the results to southern Mediterranean populations. There is currently a lack of data and results on the combined use of vegetables and red meat to mitigate the risks associated with high red meat consumption, particularly for individuals with higher meat intake.

Another important strength is the FFQ, which is the most commonly used dietary assessment method for quantifying food intake. However, it also has some limitations. The main one is the nature of the vegetables used. By increasing the numerosity and stratification of the vegetables, the protective role of vegetables in human nutrition could be further clarified.

Furthermore, it would be optimal in a future paper to consider not only individual effects (such as individual foods or physical activity), but complex patterns in order to study the complex relationships that lead to cancer development [58].

## Figures and Tables

**Figure 1 nutrients-16-01207-f001:**
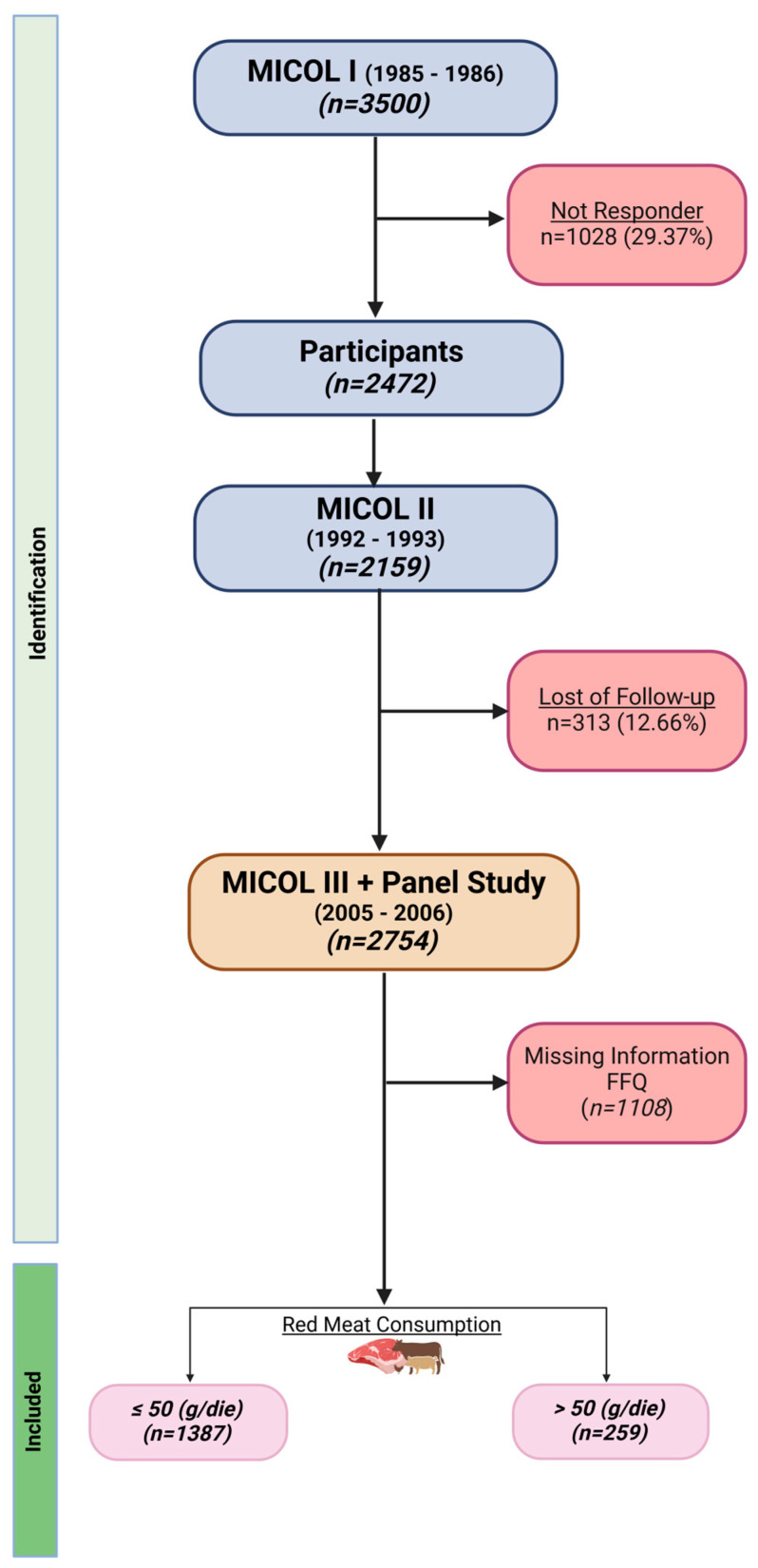
Flowchart of MICOL study. Image created with BioRender (accessed on 13 April 2024).

**Figure 2 nutrients-16-01207-f002:**
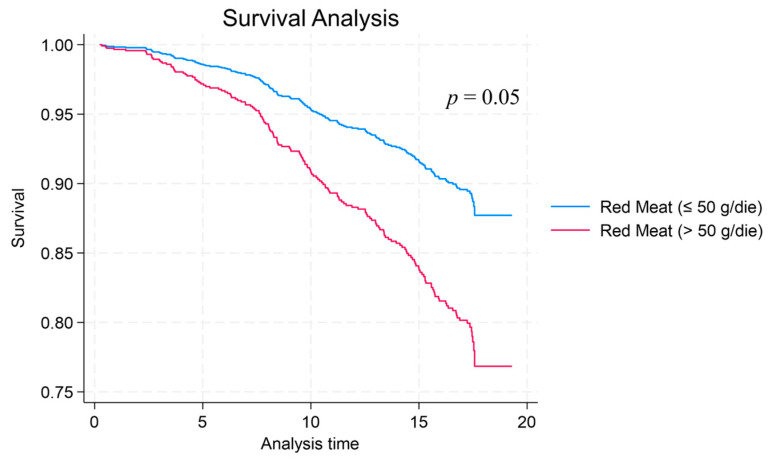
Kaplan–Meir curve comparing high versus low (>50 g/die vs. ≤50 g/die) intakes of red meat.

**Figure 3 nutrients-16-01207-f003:**
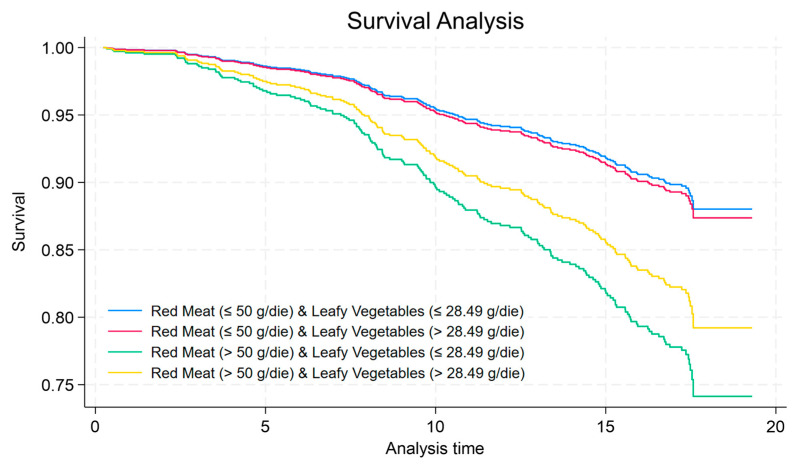
Kaplan–Meir curve comparing all combinations of red meat and leafy vegetables intakes.

**Table 1 nutrients-16-01207-t001:** Epidemiological and clinical characteristics of subjects, stratified by red meat consumption (≤50 g/die > 50 g/die). MICOL Cohort (*n* = 1646).

Parameters *	Total Cohort (*n* = 1646)	Red Meat (g/die)		*p* ^^^
		≤50 (*n* = 1387)	>50 (*n* = 259)	
Gender (M) (%)	912 (55.41)	713 (51.41)	199 (76.83)	<0.001 ^ѱ^
Age (yrs)	65.39 ± 9.47	66.06 ± 9.43	61.84 ± 8.88	<0.0001
Degree of Education (%)				0.28 ^ѱ^
No	987 (60.48)	826 (59.99)	161 (63.14)	
Elementary School	292 (17.89)	242 (17.57)	50 (19.61)	
Secondary School	205 (12.56)	176 (12.78)	29 (11.37)	
High School	64 (3.92)	59 (4.29)	5 (1.96)	
Short Degree	84 (5.15)	74 (5.37)	10 (3.92)	
Smoke (Yes) (%)	238 (14.49)	180 (13.02)	58 (22.39)	<0.001 ^ѱ^
BMI (Kg/m^2^)	29.80 ± 9.90	29.69 ± 10.47	30.38 ± 5.93	0.09
Systolic Pressure (mmHg)	138.67 ± 84.73	139.12 ± 82.62	136.30 ± 95.37	<0.0001
Diastolic Pressure (mmHg)	84.09 ± 88.29	83.71 ± 86.08	86.12 ± 99.44	0.68
Diabetes (Yes) (%)	237 (14.42)	206 (14.87)	31 (11.97)	0.22 ^ѱ^
Hypertension (Yes) (%)	808 (49.27)	707 (51.16)	101 (39.15)	<0.001 ^ѱ^
Blood Parameters				
Glucose (mg/dL)	118.47 ± 245.85	119.34 ± 267.44	113.80 ± 33.21	0.08
Cholesterol (mg/mL)	205.94 ± 244.73	206.61 ± 266.02	202.36 ± 41.07	0.62
HDL (mg/dL)	63.18 ± 346.92	58.60 ± 267.44	87.71 ± 618.38	0.01
LDL (mg/dL)	157.12 ± 596.61	142.47 ± 460.47	235.55 ± 1059.51	0.89
Triglycerides (mg/dL)	151.49 ± 357.46	142.39 ± 279.00	200.22 ± 627.46	0.002
Fatty Acids (mg/dL)	683.86 ± 2149.95	679.54 ± 2411.83	707.01 ± 2467.59	0.30
Total Bilirubin (mg/dL)	134.52 ± 1148.48	159.48 ± 1249.61	0.87 ± 0.29	0.67
GOT (U/L)	638.96 ± 2419.08	640.41 ± 2421.92	631.17 ± 2408.48	0.46
GGT (U/I)	97.39 ± 883.95	82.94 ± 801.90	174.75 ± 1233.01	0.0002
SGPT (U/L)	24.13 ± 246.67	25.12 ± 268.64	18.83 ± 14.17	0.0002
Leafy Vegetables (g/die) (%)				0.01 ^ѱ^
≤28.49	832 (50.55)	720 (51.91)	112 (43.24)	
>28.49	814 (49.45)	667 (48.09)	147 (56.76)	

* Expressed as mean and standard deviation for continuous, and as frequency and percentage (%) for categorical variables. ^^^ Wilcoxon rank-sum test (Mann–Whitney), ^ψ^ chi-squared test. Abbreviations: M, Male; BMI, body mass index; HDL, High-Density Lipoprotein; LDL, Low-Density Lipoprotein; GOT, Aspartate Amino Transferase; GGT, Gamma-Glutamyl Transferase; SGPT, Serum Glutamic Pyruvic Transaminase; *p*, *p*-value.

**Table 2 nutrients-16-01207-t002:** Cox regression model on red meat consumption and its interaction with leafy vegetable median intake, adjusted for covariates ^, inserted singly in the model.

Parameters	HR	se (HR)	*p*	95% (C.I.)
Red Meat (g/die)				
≤50 *[Ref.]*	-	-	-	-
>50	2.02	0.41	0.001	1.36 to 3.01
Red Meat # Leafy Vegetables (g/die)				
≤50 & ≤28.49 *[Ref.]*	-	-	-	-
≤50 & >28.49	1.06	0.18	0.73	0.75 to 1.49
>50 & ≤28.49	2.36	0.64	0.002	1.38 to 4.01
>50 & >28.49	1.85	0.52	0.03	1.06 to 3.22

Abbreviations: HR, Hazard Ratio; se (HR), Standard Error of HR; 95% (C.I.), 95% Confidence Interval; Ref. Reference Category; *p*, *p*-value. ^ Models adjusted for age, gender, BMI, cholesterol, triglycerides, and Kcal.

## Data Availability

The original contributions presented in this study are included in the article. Further inquiries can be directed to the corresponding author.

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
