# Peer review of "Adding a Leafy Vegetable Fraction to Diets Decreases the Risk of Red Meat Mortality in MASLD Subjects: Results from the MICOL Cohort"

_nutrients, 2024, doi:10.3390/nu16081207_

Round 1

Reviewer 1 Report

Comments and Suggestions for Authors

This research prepared by Donghia et al. aimed to investigate how leafy vegetables may protect against the negative effects of high red meat consumption. The study involved 1646 participants from the fourth recall of the MICOL study, divided into two groups based on their intake of red meat (≤ 50 grams per day and > 50 grams per day), to analyze cancer mortality. Only 15.73% of the participants consumed more than 50 grams of red meat per day. Leafy vegetable intake was categorized according to the median grams consumed per day and analyzed in conjunction with red meat intake. Findings indicate that consuming approximately 30 grams per day of leafy vegetables was associated with a reduced risk of mortality, highlighting the protective role of vegetables, particularly among subjects with high red meat intake. This study is the first to demonstrate such an association and underscores the importance of including leafy vegetables in the diet to mitigate the risks associated with high red meat consumption, particularly for individuals with higher meat intake.

The paper included enough and necessary information. The introduction provided background and information relevant to the study. The methods should be more described. The results are clear and replicable. The Authors have presented sufficient data.  The article is easy to read and logically structured. The obtained results are presented in the form of figures and tables, which are  easy to understand; the discussion- supports the results properly and appropriately refers to the current literature; the conclusions- based on the obtained results

The remarks, which should be included:

1.     In section 2.1. please add the diagram flow examined patients.

2.     The methods should be more described because it is difficult to understand the study.

3.     Please add the limitations and strengths of your study

4.      Please add practical recommendations for patients.

Comments on the Quality of English Language

Some minor correction is needed

Author Response

The revisions are attached in a .pdf file.

Reviewer 2 Report

Comments and Suggestions for Authors

Report on the manuscript nutrients-2938492 entitled: Adding a leafy vegetables fraction decreases the risk of red meat mortality in MASLD subjects: results from the MICOL cohort.

There is a strong opinion that epidemiological studies that consider mortality should be read with caution. In fact, these studies should be written with a stricter scientific rigor because the future science-industry-public transference might depend on them.

Before being considered for publication, several concerns/doubts/comments need to be addressed.

-          L. 28-43. These comments are too general and scientifically unprecise. In fact, there is available an update (more current publication) of Ref. 1 that should be considered:

Clinton, S.K.; Giovannucci, E.L.; Hursting, S.D. The World Cancer Research Fund/American Institute for Cancer Research Third Expert Report on Diet, Nutrition, Physical Activity, and Cancer: Impact and Future Directions. J. Nutr. 2020, 150, 663-671, doi:10.1093/jn/nxz268.

In addition, the statement:

“It is worth emphasising the importance of considering how different overall patterns of diet and physical activity combine to create a metabolic state that is more, or less, conducive to the development of cancer…”

should be considered.

https://www.wcrf.org/wp-content/uploads/2021/02/Summary-of-Third-Expert-Report-2018.pdf

-          L. 48. Reference 8? What about references 4 to 7?

-          L. 56-61. These lines somehow contradict the beginning of the section (L. 28-43), thus supporting my first comment.

-          L. 101-103 and Table S1: The table needs to be improved: “liver” is not a “red meat” (liver from?). “salad” is not a “vegetable”.
Where were these data read from? Please, rewrite and add a reference.

-          L. 104-108. Table S2 data…but implications/relationship? The data/results need to be described and the implication/relationship with the present study stated.

-          Table 2 (regression models) together with Table 1:

It was described as the covariables: age, gender, BMI, cholesterol, triglycerides, and Kcal.

Nevertheless, in Table 1, smoke together with other parameters were statistically significant. How were these effects considered in the study?

Please, improve both the Results and the Discussion sections.

-          L. 253-256. Please, rewrite the paragraph. The study was carried out in fecal water but “L. 254. Increased fecal water” is unprecise and must be improved.

-          L. 261-264. Please, add the proper reference/cite and improve the description of the referenced study and the relationship with the author’s data/results.

-          L. 265-275. Please, add the proper reference/cite and improve the description of the referenced study and the relationship with the author’s data/results.

-          L. 312-314. This statement needs to be included in the Discussion section. In fact, the use of references in the Conclusions is strongly discouraged.

-          The authors have recently published several papers related to the aim of the current study.

1.          Bianco, A.; Russo, F.; Franco, I.; Riezzo, G.; Donghia, R.; Curci, R.; Bonfiglio, C.; Prospero, L.; D’Attoma, B.; Ignazzi, A., et al. Enhanced Physical Capacity and Gastrointestinal Symptom Improvement in Southern Italian IBS Patients following Three Months of Moderate Aerobic Exercise. Journal of Clinical Medicine 2023, 12, doi:10.3390/jcm12216786.

2.          Bonfiglio, C.; Cuccaro, F.; Campanella, A.; Rosso, N.; Tatoli, R.; Giannelli, G.; Donghia, R. Effect of Intake of Extra Virgin Olive Oil on Mortality in a South Italian Cohort with and without NAFLD. Nutrients 2023, 15, doi:10.3390/nu15214593.

3.          Donghia, R.; Guerra, V.; Pesole, P.L.; Liso, M. Contribution of macro- and micronutrients intake to gastrointestinal cancer mortality in the ONCONUT cohort: Classical vs. modern approaches. Frontiers in Nutrition 2023, 10, doi:10.3389/fnut.2023.1066749.

4.          Donghia, R.; Pesole, P.L.; Castellaneta, A.; Coletta, S.; Squeo, F.; Bonfiglio, C.; De Pergola, G.; Rinaldi, R.; De Nucci, S.; Giannelli, G., et al. Age-Related Dietary Habits and Blood Biochemical Parameters in Patients with and without Steatosis—MICOL Cohort. Nutrients 2023, 15, doi:10.3390/nu15184058.

5.          Donghia, R.; Pesole, P.L.; Coletta, S.; Bonfiglio, C.; De Pergola, G.; De Nucci, S.; Rinaldi, R.; Giannelli, G. Food Network Analysis in Non-Obese Patients with or without Steatosis. Nutrients 2023, 15, doi:10.3390/nu15122713.

6.          Donghia, R.; Schiano Di Cola, R.; Cesaro, F.; Vitale, A.; Lippolis, G.; Lisco, T.; Isernia, R.; De Pergola, G.; De Nucci, S.; Rinaldi, R., et al. Gender and Liver Steatosis Discriminate Different Physiological Patterns in Obese Patients Undergoing Bariatric Surgery: Obesity Center Cohort. Nutrients 2023, 15, doi:10.3390/nu15102381.

7.          Lampignano, L.; Tatoli, R.; Donghia, R.; Bortone, I.; Castellana, F.; Zupo, R.; Lozupone, M.; Panza, F.; Conte, C.; Sardone, R. Nutritional patterns as machine learning predictors of liver health in a population of elderly subjects. Nutrition, Metabolism and Cardiovascular Diseases 2023, 33, 2233-2241, doi:10.1016/j.numecd.2023.07.009.

8.          Zupo, R.; Donghia, R.; Castellana, F.; Bortone, I.; De Nucci, S.; Sila, A.; Tatoli, R.; Lampignano, L.; Sborgia, G.; Panza, F., et al. Ultra-processed food consumption and nutritional frailty in older age. GeroScience 2023, 45, 2229-2243, doi:10.1007/s11357-023-00753-1.

9.          Bonfiglio, C.; Campanella, A.; Donghia, R.; Bianco, A.; Franco, I.; Curci, R.; Bagnato, C.B.; Tatoli, R.; Giannelli, G.; Cuccaro, F. Development and Internal Validation of a Model for Predicting Overall Survival in Subjects with MAFLD: A Cohort Study. Journal of Clinical Medicine 2024, 13, doi:10.3390/jcm13041181.

10.        Campanella, A.; Bonfiglio, C.; Cuccaro, F.; Donghia, R.; Tatoli, R.; Giannelli, G. High Adherence to a Mediterranean Alcohol-Drinking Pattern and Mediterranean Diet Can Mitigate the Harmful Effect of Alcohol on Mortality Risk. Nutrients 2024, 16, doi:10.3390/nu16010059.

11.        Donghia, R.; Campanella, A.; Bonfiglio, C.; Cuccaro, F.; Tatoli, R.; Giannelli, G. Protective Role of Lycopene in Subjects with Liver Disease: NUTRIHEP Study. Nutrients 2024, 16, doi:10.3390/nu16040562.

12.        Tatoli, R.; Bonfiglio, C.; Cuccaro, F.; Campanella, A.; Coletta, S.; Pesole, P.L.; Giannelli, G.; Donghia, R. Effects of Egg Consumption on Subjects with SLD or Hypertension: A MICOL Study. Nutrients 2024, 16, doi:10.3390/nu16030430.

The discussion should include previous results and their relationship and effect with the new ones as the available knowledge grows. In fact, the authors published an effect of olive oil and the physical exercise. However, these 2 effects have not even been mentioned in the discussion.

-          The authors have recently published several papers in the same journal:

1. Bonfiglio, C. et al. Effect of Intake of Extra Virgin Olive Oil on Mortality in a South Italian Cohort with and without NAFLD. Nutrients 2023, 15, doi:10.3390/nu15214593.

2. Campanella, A. et al. High Adherence to a Mediterranean Alcohol-Drinking Pattern and Mediterranean Diet Can Mitigate the Harmful Effect of Alcohol on Mortality Risk. Nutrients 2024, 16, doi:10.3390/nu16010059.

3. Donghia, R. et al. Protective Role of Lycopene in Subjects with Liver Disease: NUTRIHEP Study. Nutrients 2024, 16, doi:10.3390/nu16040562.

4. Donghia, R. et al. Age-Related Dietary Habits and Blood Biochemical Parameters in Patients with and without Steatosis—MICOL Cohort. Nutrients 2023, 15, doi:10.3390/nu15184058.

5. Donghia, R. et al. Food Network Analysis in Non-Obese Patients with or without Steatosis. Nutrients 2023, 15, doi:10.3390/nu15122713.

6. Donghia, R. et al. Gender and Liver Steatosis Discriminate Different Physiological Patterns in Obese Patients Undergoing Bariatric Surgery: Obesity Center Cohort. Nutrients 2023, 15, doi:10.3390/nu15102381.

7. Tatoli, R. et al. Effects of Egg Consumption on Subjects with SLD or Hypertension: A MICOL Study. Nutrients 2024, 16, doi:10.3390/nu16030430.

Nevertheless, they seem to have forgotten the journal requirements regarding citations and references.

Please, review the Reference list. The format does not follow the journal requirements.

Author Response

(The authors gave the same response as above.)

Reviewer 3 Report

Comments and Suggestions for Authors

STRUCTURE

-       The manuscript is correctly structured, but the tables are not aligned to the centre. Please check this

o   https://www.mdpi.com/journal/obesities/instructions

TITLE AND ABSTRACT

-       Title: The title of your manuscript is concise, specific and relevant.

INTRODUCTION

-       The content of the introduction is adequate, however, when you go to the references cited, you notice that only 2 are less than 5 years old. The citations are too outdated. It is recommended that the authors insert more up-to-date studies.

-       The introduction contains the objective of the study, but what about the hypothesis? the authors should make explicit what their research hypothesis is.

-       The authors are recommended to do some more research on red meat consumption. Is fresh meat the same as processed meat? Processed foods are the ones that mostly end up harming consumers. And it seems that all red meat is always lumped into one group. When we know that a pork fillet is not the same as a piece of bacon or sausage.

o   Zhang R, Zhang H, Wang Y, Tang LJ, Li G, Huang OY, Chen SD, Targher G, Byrne CD, Gu BB, Zheng MH. Higher consumption of animal organ meat is associated with a lower prevalence of nonalcoholic steatohepatitis. Hepatobiliary Surg Nutr. 2023 Oct 1;12(5):645-657. doi: 10.21037/hbsn-21-468. Epub 2022 Jun 16. PMID: 37886189; PMCID: PMC10598295.

o   Recaredo G, Marin-Alejandre BA, Cantero I, Monreal JI, Herrero JI, Benito-Boillos A, Elorz M, Tur JA, Martínez JA, Zulet MA, Abete I. Association between Different Animal Protein Sources and Liver Status in Obese Subjects with Non-Alcoholic Fatty Liver Disease: Fatty Liver in Obesity (FLiO) Study. Nutrients. 2019 Oct 3;11(10):2359. doi: 10.3390/nu11102359. PMID: 31623368; PMCID: PMC6836147.

MATERIAL AND METHODS

-       Line 91: “Methodological details of this population-based study have been described in previously published papers”. Please mention again the methodology used for the selection of the sample.

-       It is recommended to add a flow chart of the sample and its follow-up.

-       What protocol was followed for blood sampling? What tools were used?

-       In addition, the biomarkers used are not indicated. However, the table does show each of them. It is important that everything is detailed. An example is shown:

-       Triglycerides. Blood samples were drawn and triglycerides were measured using an Accutrend® Plus. The Accutrend ® Plus test uses capillary serum and relies on the retention of blood cells by filtration through glass fibre when a drop of blood is applied to the test strip. An enzymatic reaction occurs on the strip when exposed to oxygen, resulting in a colour. The reflectance of the strip is measured at 660 nm and the triglyceride concentration is calculated using a simple algorithm. The accuracy of the Accutrend® Plus, as stated in the product information, is 3.4% [1].

o   Scafoglieri A, Tresignie J, Provyn S, Clarys JP, Bautmans I. Reproducibility, accuracy and concordance of Accutrend Plus for measuring circulating lipid concentration in adults. Biochem Med (Zagreb). 2012;22(1):100-8. doi: 10.11613/bm.2012.011. PMID: 22384524; PMCID: PMC4062330.

2.5. Statistics analysis

-       The authors describe the statistical methods used to compare the primary and secondary outcome groups, as well as methods for further analysis, such as subgroup analysis and adjusted analysis.

RESULTS

-       Table 1. Not explained what is "P-value". Applicable to all other tables.

DISCUSSION

-       The limitations of the study are included, but conservative overall interpretation of the results has been provided taking into account the objectives, the multiplicity of analyses, the results of similar studies and other relevant evidence.

-       Line 261: what does crc mean?

-       The discussion is too brief, it does not provide sufficient arguments in relation to other similar and current research. You can compare with other research and/or systematic reviews:

o   Bradley C. Johnston, Dena Zeraatkar, Mi Ah Han, et al. Unprocessed Red Meat and Processed Meat Consumption: Dietary Guideline Recommendations From the Nutritional Recommendations (NutriRECS) Consortium. Ann Intern Med.2019;171:756-764. [Epub 1 October 2019]. doi:10.7326/M19-1621

-       On the other hand, it is also recommended to emphasise physical exercise. It is also interesting to know the effect of movement on metabolism. But above all, mention specifically which types of red meat (food) the authors cited refer to.

CONCLUSION

-       Conclusion: has been included. In this section, researchers provide a brief conclusion.

-       Line 307: “In conclusion, in the present study we observed a significant association between the consumption…” Do not use first person plural. Change to third person. Applicable to the rest of the document.

REFERENCES

-       References are numbered in order of appearance in the text. They are correctly placed in square brackets [ ] and placed before punctuation.

-       Check references to ensure they are in line with the text.

o   https://www.mdpi.com/journal/obesities/instructions

-       References should be described as follows, depending on the type of work. For example, reference 36 and 37 are wrong. There are others that are the same, check them all.

o   Journal Articles:

1.     Author 1, A.B.; Author 2, C.D. Title of the article. Abbreviated Journal Name Year, Volume, page range.

36. Li H.; Wang X., Ye M.; Zhang S., Zhang Q., Meng G.; Liu L.; Wu H.; Gu Y.; Wang Y.; Zhang T.; Sun S.; Wang X., Zhou M.; Jia Q., Song K.; Wang Y.; Niu K. Does a high intake of green leafy vegetables protect from NAFLD? Evidence from a large population study. Nutr Metab Cardiovasc Dis 2021, 31, 1691-1701.

37. Guo W., Ge X., Lu J.; Xu X., Gao J.; Wang Q., Song C.; Zhang Q.; Yu C. Diet and Risk of Non-Alcoholic Fatty Liver Disease, Cirrhosis, and Liver Cancer: A Large Prospective Cohort Study in UK Biobank. Nutrients 2022, 14, 5335.

-       A bibliographic manager should be used so that the citations are well inserted, homogeneous and in accordance with the standards recommended by the journal. The name of the journal has to be abbreviated.

Author Response

(The authors gave the same response as above.)

Round 2

Reviewer 2 Report

Comments and Suggestions for Authors

The manuscript has been improved but not all the issues/comments have been addressed properly by the authors.

-          The authors indicate that Supplementary Tables 1 and 2 serve to describe the execution of the study.  The usefulness of these tables is somewhat questionable, as they are not described, commented on, or discussed anywhere.

-          As mentioned in the previous revision, in Table 2 (regression models) together with Table 1:

It was described as the covariables: age, gender, BMI, cholesterol, triglycerides, and Kcal.

Nevertheless, in Table 1, smoke together with other parameters were statistically significant.

The authors responded that: Table 1 was a comparison table, while Table 2 evaluated the specific mortality risk. The use of those variables to adjust the model was precisely to remove their effect normally associated with mortality and therefore study the relationship between meat consumption and specific mortality "at the same level". This approach was reported in the statistical analysis section and in the end note of table 2.

Nevertheless, the issue regarding “smoke” or “hypertension”, etc. has not been addressed.

Smoke, hypertension, etc. were statistically significant (table 2) and are widely described as mortality risks. Then, why were not considered as covariables? Could the use of leafy vegetables surpass the effect of smoking or hypertension?

-          L.276-362. Please, review and improve the Discussion section. It seems a literature review more than a research discussion.

-          Please, review the whole manuscript and use “g” instead of “gr” when “grams” is meant.

-          L. 95. Please, delete one “was”.

-          L. 103. Remove the “7”.

-          L. 158. “for covariates” or “for covariables”?

-          L. 163-164. Double-check the use of parentheses.

-          L. 297 and 300. Review the citation format.

-          Please, review the Reference list. The format does not follow the journal requirements yet.

Author Response

Requests are attached as .pdf files.
